# POINTHR: EXPLORING HIGH-RESOLUTION ARCHITECTURES FOR 3D POINT CLOUD SEGMENTATION

## ABSTRACT

Significant progress has been made recently in point cloud segmentation utilizing an encoder-decoder framework, which initially encodes point clouds into low-resolution representations and subsequently decodes high-resolution predictions. Inspired by the success of high-resolution architectures in image dense prediction, which always maintains a high-resolution representation throughout the entire learning process, we consider it also highly important for 3D dense point cloud analysis. Therefore, in this paper, we explore high-resolution architectures for 3D point cloud segmentation. Specifically, we generalize high-resolution architectures using a unified pipeline named PointHR, which includes a knn-based sequence operator for feature extraction and a differential resampling operator to efficiently communicate different resolutions. Additionally, we propose to avoid numerous on-the-fly computations of high-resolution architectures by *pre-computing* the indices for both sequence and resampling operators. By doing so, we deliver highly competitive high-resolution architectures while capitalizing on the benefits of well-designed point cloud blocks without additional effort. To evaluate these architectures for dense point cloud analysis, we conduct thorough experiments using S3DIS and ScanNetV2 datasets, where the proposed PointHR outperforms recent state-of-the-art methods without any bells and whistles. The source code is available in the supplementary material and will be made publicly accessible.

## 1 INTRODUCTION

In recent years, 3D point clouds have gained extensive attention due to their crucial role in many real-world applications, such as autonomous driving (Aksoy et al., 2020; Li et al., 2020), robotics (Li et al., 2019; Yang et al., 2020), and AR/VR (Alexiou et al., 2020; Chen et al., 2019). Unlike 2D images, each point cloud consists of a set of 3D points characterized by their Cartesian coordinates $(x, y, z)$, providing a view-invariant and geometry-accurate representation of the real-world 3D scene. 3D point cloud segmentation aims to predict semantic labels for all points of the point cloud, which requires a learned representation that is both spatially accurate (for individual points) and semantically rich (for category prediction). However, developing effective deep architectures for 3D point cloud representation learning is non-trivial.

3D scenes represented by point clouds often contain objects of different scales, such as a small cup on a large table in an office room, which requires the deep model to capture multi-scale contexts within point clouds. In typical deep neural networks (Chen et al., 2017; He et al., 2016; Long et al., 2015), we tend to believe that shallow features (high resolution) contain more accurate spatial information while deep features (low resolution) include more semantic clues. Therefore, previous point cloud segmentation methods (Qian et al., 2022; Zhao et al., 2021) mainly explore multi-scale information by downsampling and upsampling features in series using an encoder-decoder paradigm: they first encode the input point clouds by progressively downsampling the point features and then decode back to the original scale using upsampling on lower scale features to generate dense predictions. Specifically, feature interactions only occur in adjacent scale representations, limiting the learning of rich multi-scale semantics. Additionally, the final largest resolution representation is recovered step-by-step from low resolution, which compromises spatial accuracy. These designs make such a vanilla encoder-decoder framework insufficient to capture rich multi-scale contexts. Inspired by high-resolution architectures for 2D visual recognition (Wang et al., 2020), we introduce PointHR, which explicitly maintains high-low resolution features in parallel during the entire network for point

cloud segmentation. Unlike previous hierarchical methods (Qian et al., 2022; Zhao et al., 2021) that use only one resolution feature in each stage, PointHR keeps multiple resolutions (from high to low) simultaneously and facilitates frequent communication between all resolutions within each stage.

Different from structurally-located pixels in 2D images, point clouds consist of a set of irregular and unordered points, making it non-trivial to employ high-resolution architectures. Therefore, we approach point clouds for high-resolution architectures as follows. Firstly, the input point clouds are considered as a sequence of $(x, y, z)$ points, allowing for a general sequence operator to be used for feature extraction. For example, a popular sequence operator could be self-attention (Vaswani et al., 2017), which was originally designed to model relationships between a sequence of text tokens. However, since the self-attention operator has quadratic time complexity $\mathcal{O}(N^2)$ for a sequence of length $N$, it is computationally infeasible to directly apply this operator to a point cloud with tens of thousands of points. One solution is to constrain the self-attention operator to perform only on each point and its $K$ nearest neighboring points, thus reducing the complexity to be linear with respect to length $N$ as demonstrated in Wu et al. (2022); Zhao et al. (2021). Another sequence operator could be pure MLPs, which can process unordered sequences when feeding permuted training data. To aggregate local information, $K$ nearest neighbor features are retrieved, followed by MLPs to fuse and update the current point feature as shown in Lin et al. (2023). Thus, we formulate the basic block as a knn-based sequence operator, allowing for the use of well-designed point cloud blocks (Lin et al., 2023; Wu et al., 2022; Zhao et al., 2021) in high-resolution architectures without additional effort.

In addition to the sequence operator, another important aspect of high-resolution architectures is the resampling operator, which can efficiently communicate different scale features in high frequency with upsampling and downsampling. A common resampling strategy in point clouds is a combination of farthest point sampling (Eldar et al., 1997; Qi et al., 2017b) with knn features aggregation/interpolation. Recently, an efficient grid-based pooling and unpooling strategy (Wu et al., 2022) has been introduced, which first splits a point cloud into non-overlapping grids and then maps each grid of points to a new one and vice versa. With the unified aforementioned formulation for sequence operators, these resampling methods can be easily adopted in PointHR. However, all of them require calculating the indices for knn collection and resampling in each operation. This incurs a significant computational cost, particularly when considering the numerous resampling operations in high-resolution architectures. Fortunately, we have found that the calculations of these indices only depend on scale, specifically the corresponding point coordinates. These coordinates remain unchanged throughout the entire network. Therefore, we propose to *pre-compute* the indices for knn collection and resampling operation, which are saved to the cache so that indices retrieval is only needed instead of on-the-fly re-computation.

Our main contributions are summarized as follows:

- We present a new framework for point cloud segmentation, PointHR, which aims to maintain high resolutions for learning both semantically-rich and spatially-precise point cloud representations.

- We explore high-resolution architectures using unified sequence and resampling operators, allowing off-the-shelf point cloud blocks and layers to be employed in PointHR without additional efforts. Besides, we *pre-compute* the indices for knn collection and resampling operation to avoid on-the-fly re-computation.

- We conduct comprehensive experiments on two popular point cloud segmentation datasets, namely S3DIS (Armeni et al., 2016) and ScanNetV2 (Dai et al., 2017), where the proposed PointHR demonstrates new state-of-the-art performance, suggesting the effectiveness of exploring high-resolution architectures for point cloud segmentation.

## 2 RELATED WORK

**High-Resolution Architectures.** High-Resolution Network (HRNet), originally proposed by Sun et al. (2019) for human pose estimation, maintains multiple branches for multi-scale features, particularly high-resolution representations, which facilitate learning spatially more precise heatmaps for pose estimation. With its repeated cross-scale feature interactions in deep layers, HRNet also learns rich semantic features. Consequently, Wang et al. (2020) extended HRNet to general dense prediction tasks such as semantic segmentation and object detection, achieving impressive performances.

HRFormer (Yuan et al., 2021) employs emerging attention blocks (Vaswani et al., 2017) to replace vanilla convolution layers in the high-resolution framework, thereby expanding its modeling capacity. Additionally, Zhang et al. (2021) adopts HRNet for person re-identification to address the issue of multiple resolutions of input images. The aforementioned studies have all focused on 2D images, and we further explore high-resolution architectures for 3D point clouds.

**Point Cloud Segmentation.** Here we mainly introduce methods that directly take raw points as input without extra transformations (*e.g.*, voxelization and projection). These point-based methods usually develop novel modules or frameworks, such as MLPs (Ma et al., 2022; Qian et al., 2022; Qi et al., 2017a;b), point convolutions (Thomas et al., 2019; Wu et al., 2019), and attentions (Guo et al., 2021; Lai et al., 2022; Park et al., 2022; Qiu et al., 2023; Zhao et al., 2021), to directly learn from raw points. Particularly, PointNet (Qi et al., 2017a) was the first to process point clouds using multi-layer perceptrons (MLPs). PointNet++ (Qi et al., 2017b) improved upon this by introducing a hierarchical neural network that learns local features. PointNext (Qian et al., 2022), on the other hand, proposes advanced training strategies to significantly improve the performance of PointNet++. Additionally, it introduces InvResMLP blocks and formulates the PointNext architecture for further improvement. Meanwhile, KPConv (Thomas et al., 2019) presents kernel point convolution as a new point convolution operator that takes neighboring points as input and processes them with spatially located weights. Recently, the popular transformer architecture has been introduced to point cloud tasks (Lai et al., 2022; Park et al., 2022; Wu et al., 2022; Zhao et al., 2021). PTv1 (Zhao et al., 2021) proposes vector attention to aggregate neighbor features. PTv2 (Wu et al., 2022) further introduces grouped vector attention to more efficiently learn discriminative representations while avoiding overfitting. Stratified Transformer (Lai et al., 2022) employs local window-based self-attention and captures long-range contexts by sampling nearby points densely and distant points sparsely. Differently, in this paper, we mainly explore how to form an effective backbone framework, and the block designs of the above methods can be trivially integrated in our framework.

## 3 METHOD

First, we present an overview of high-resolution architectures for 3D point clouds. Next, we delve into the unified format of sequence operator with various instantiations. Finally, we investigate efficient multi-scale fusion in conjunction with the resampling operator using the *pre-compute* strategy.

### 3.1 HIGH-RESOLUTION ARCHITECTURES

A point cloud can be represented as a sequence of $N$ points $\boldsymbol{P} \in \mathbb{R}^{N \times C_{raw}}$, where $C_{raw}$ is the feature dimension of each point that typically includes xyz coordinates as well as other attributes such as its normal vector. The goal of point cloud segmentation is to predict a semantic category label for each point $(x, y, z) \in \boldsymbol{P}$. Specifically, the final predictions share the same spatial dimension with the raw points, *i.e.*, $\boldsymbol{Y} \in R^{N, cls}$, where $cls$ is the total number of semantic categories.

The overall PointHR pipeline for point cloud segmentation is illustrated in Figure 1, where the raw input point clouds $N \times C_{raw}$ is first embedded to $N \times C_0$, and then downsampled to $N/S \times C_1$. Next, it starts with a high-resolution branch as the first stage by taking the point feature $N/S \times C_1$ as input. Subsequently, additional high-to-low resolution branches calculated by spatially dividing the factor $S$ and doubling the channel of feature dimension are incrementally integrated into the architecture as new stages. For $i = 1, 2, 3, 4$, the $i_{th}$ stage consisting of $i$ branches outputs $i$ different scales point features, and each branch has $M_i$ blocks that is building by stacking $B_i$ sequence operators. After each stage, the learned multi-resolution features are fused as the input of the next stage in a per-branch manner. Notably, the resolution corresponds to the number of points when applying high-resolution architectures for point clouds. The entire framework can be formulated as follows:

$$
\begin{array}{ccccccc}
\boldsymbol{F}_{11} & \to & \boldsymbol{F}_{21} & \to & \boldsymbol{F}_{31} & \to & \boldsymbol{F}_{41} \\
& \searrow & \boldsymbol{F}_{22} & \to & \boldsymbol{F}_{32} & \to & \boldsymbol{F}_{42} \\
& & & \searrow & \boldsymbol{F}_{33} & \to & \boldsymbol{F}_{43} \\
& & & & & \searrow & \boldsymbol{F}_{44},
\end{array}
\tag{1}
$$

where point feature $\boldsymbol{F}_{ij} \in \mathbb{R}^{\frac{N}{S^i} \times 2^{j-1} C_i}$ where $i \in \{1, 2, 3, 4\}$ and $j \in \{1, \cdots, i\}$. It should be also noted that previous hierarchical methods (Qian et al., 2022; Wu et al., 2022) only contain $\boldsymbol{F}_{11} \to \boldsymbol{F}_{22} \to \boldsymbol{F}_{33} \to \boldsymbol{F}_{44}$, which is corresponding to the red-arrow flow in Figure 1.

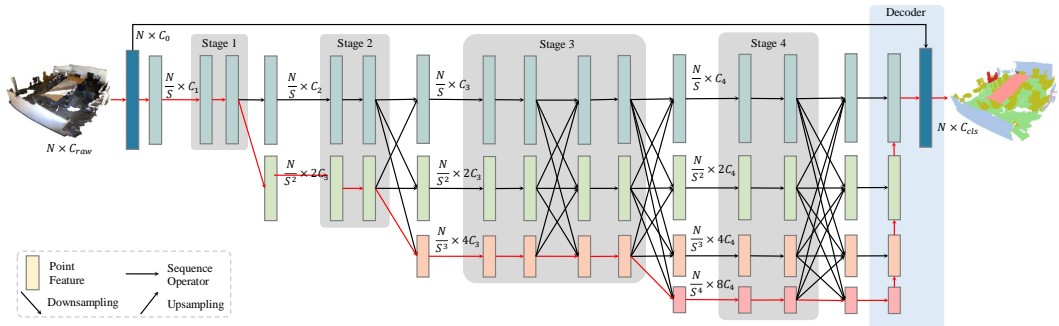

Figure 1: The overall PointHR pipeline for point cloud segmentation.

These outputs from the final stage of PointHR join forces with the original resolution feature in the decoder, enabling the propagation of information from low-resolution to high-resolution. Finally, a segmentation head that consists of linear layers generates the final prediction of $N \times C_{cls}$, where $C_{cls}$ represents the number of semantic categories. We summarize the typical configurations of PointHR via the number of modules $M_i$, the number of blocks $B_i$, and the number of channels $C_i$, *e.g.*, $(M_1, M_2, M_3, M_4) = (1, 1, 2, 1)$ and $(B_1, B_2, B_3, B_4) = (2, 2, 2, 2)$ corresponding to Figure 1. A detailed formulation of PointHR configurations is shown in Appendix A.

## 3.2 SEQUENCE OPERATOR

For 2D images, high-resolution architectures (Wang et al., 2020) usually adopt convolutional layers as the basic blocks to extract local information, while it is not straightforward to employ such an operation on point clouds due to the irregular and unordered characteristics. Therefore, to explore high-resolution architectures for point clouds, we first formulate the aforementioned basic block via a unified sequence operator, *i.e.*, regarding point clouds as a sequence of points and treating it as a sequence processing task. By doing this, we hope that most existing point cloud blocks can be directly used in high-resolution architectures. Specifically, when taking the point clouds $\boldsymbol{P} \in \mathbb{R}^{N \times C}$ as input, the sequence operator $\Theta$ will first embed each point $\boldsymbol{p}_i$ where $i \in \{1, \cdots, N\}$ into new feature space $\bar{\boldsymbol{p}}_i$, then $K$ nearest neighbors (KNN) for each point are fetched as $\bar{\boldsymbol{p}}_j$ where $j \in \{1, \cdots, K\}$ to collect local clues. After that, the sequence operator aggregates those local information to the current point $\hat{\boldsymbol{p}}_i$. Lastly, the final representation $\tilde{\boldsymbol{p}}_i$ is obtained by incorporating original features with updated current features. The whole process is illustrated in the left part of Figure 2 and can be mathematically formulated as follows:

$$\bar{\boldsymbol{p}}_i = \Theta_{se}(\boldsymbol{p}_i) \quad \Rightarrow \quad \bar{\boldsymbol{p}}_j = \Theta_{nf}(\bar{\boldsymbol{p}}_i) \tag{2}$$

$$\hat{\boldsymbol{p}}_i = \Theta_{na}(\bar{\boldsymbol{p}}_j) \quad \Rightarrow \quad \tilde{\boldsymbol{p}}_i = \Theta_{su}(\hat{\boldsymbol{p}}_i) + \boldsymbol{p}_i. \tag{3}$$

where $\Theta_{se}, \Theta_{nf}, \Theta_{na}$, and $\Theta_{su}$ indicate *Sequence Embed, Neighbors Fetch, Neighbors Aggregation*, and *Sequence Update*, respectively, in Figure 2.

To show the effectiveness of the aforementioned sequence operator formulation, we discuss several popular point cloud blocks as the possible instantialization in the following. For example, self-attention (Vaswani et al., 2017) proposed for handling a sequence of words is capable of capturing the relationships of all elements in the sequence. Previous approaches (Wu et al., 2022; Zhao et al., 2021) employ the attention only on each point with its $K$ neighboring points to reduce time complexity from $\mathcal{O}(N^2)$ to $\mathcal{O}(NK^2)$, rather than utilizing a global attention on all sequence elements. In particular, PTv1 (Zhao et al., 2021) has shown that vector attention is more effective in handling point clouds than the original scalar attention (Vaswani et al., 2017). It employs attention weights that are vectors calculated based on the relation operation between query and key, which can effectively modulate individual feature channels. Specifically, given a point $\boldsymbol{p}_i$ and its neighbors $\mathcal{N}(\boldsymbol{p}_i) = \{\boldsymbol{p}_j | \boldsymbol{p}_j \in \text{KNN}(\boldsymbol{p}_i)\}$, Multilayer perceptrons (MLPs) are employed to map the point feature to query $\boldsymbol{q}_i$, key $\boldsymbol{k}_i$, and value $\boldsymbol{v}_i$, and then the vector attention can be formulated as follows:

$$\boldsymbol{w}_{ij} = \varphi(\sigma(\boldsymbol{q}_i, \boldsymbol{k}_i)), \qquad \boldsymbol{p}_i^{attn} = \sum_{\boldsymbol{p}_j \in \mathcal{N}(\boldsymbol{p}_i)} \text{Softmax}(\boldsymbol{W}_i)_j \odot \boldsymbol{v}_j, \tag{4}$$

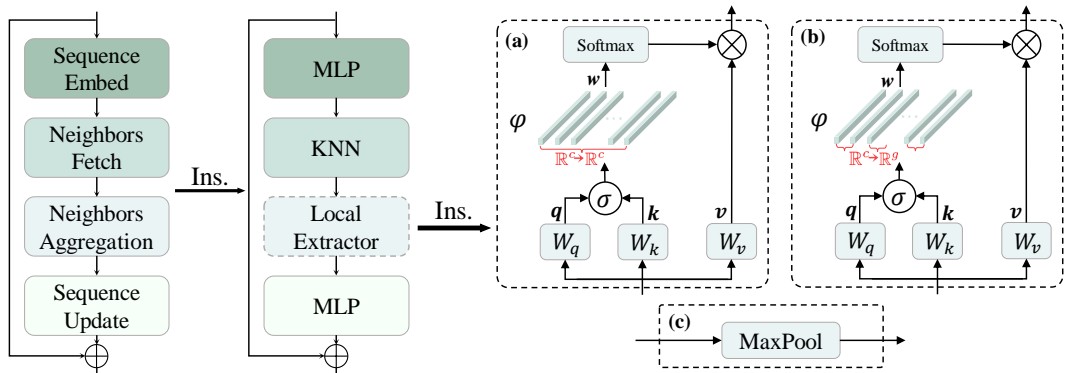

Figure 2: An illustration of the general sequence operator $\Theta$ along with its instantialization. Note that (a) vector attention (Zhao et al., 2021), (b) grouped vector attention (Wu et al., 2022), and (c) pure MLPs (Lin et al., 2023) are three instantializations of the Local Extractor defined by Equation 4, 5, and 6 respectively.

where $\odot$ represents the Hadamard product. $\sigma$ denotes a relation operation such as subtraction. $\varphi : \mathbb{R}^c \rightarrow \mathbb{R}^c$ is the MLPs which calculates attention vectors to re-weight $\boldsymbol{v}_j$ by channels before performing aggregation. PTv2 (Wu et al., 2022) further introduces a grouped vector attention to improve the model efficiency. Specifically, it is achieved by dividing channels of the value $\boldsymbol{v}_i \in \mathbb{R}^c$ evenly into $g$ groups ($1 \leq g \leq c$). Then MLPs $\varphi : \mathbb{R}^c \rightarrow \mathbb{R}^g$ outputs a grouped attention vector with $g$ channels instead of $c$. Channels within the same group share the same scalar attention weight from the grouped attention vector. As a result, Equation 4 is modified as follows:

$$\boldsymbol{w}_{ij} = \varphi(\sigma(\boldsymbol{q}_i, \boldsymbol{k}_i)), \qquad \boldsymbol{p}_i^{attn} = \sum_{\boldsymbol{p}_j \in \mathcal{N}(\boldsymbol{p}_i)} \sum_{l=1}^{g} \sum_{m=1}^{c/g} \text{Softmax}(\boldsymbol{W}_i)_{jl} \cdot \boldsymbol{v}_j^{lc/g+m}. \tag{5}$$

The sequence operator can be pure MLPs as well, which can handle unordered sequences when augmenting the training data by all kinds of permutations. Specifically, in Lin et al. (2023), the features of $K$ nearest neighbours are used to collect local clues, which is followed by maxpooling and MLPs to fuse current point features, which can be formulated as follows:

$$\boldsymbol{p}_i^{aggre} = \text{MaxPool}(\phi(\boldsymbol{p}_j)), \qquad \boldsymbol{p}_j \in \mathcal{N}(\boldsymbol{p}_i), \tag{6}$$

where $\phi$ is the MLPs that embeds the input point feature $\boldsymbol{p}_i$, and $\mathcal{N}(\boldsymbol{p}_i)$ represents the neighbors of $\boldsymbol{p}_i$. From an united perspective, we can consider the instantialization of Equation 2 and 3 as follows:

$$\Theta_{se} = \text{MLP}, \quad \Theta_{nf} = \text{KNN}, \quad \Theta_{su} = \text{MLP}. \tag{7}$$

Notably, the only difference lies in $\Theta_{na}$, which acts as the local extractor to capture neighbor information as shown in the right part of Figure 2, defined by Equation 4, 5, and 6.

### 3.3 RESAMPLING OPERATOR AND MULTI-SCALE FUSION

Besides the sequence operator, a resampling operator is required to perform cross-resolution interactions between different branches. Specifically, for $j_{th}$ branch where $j \in \{1, \cdots, i\}$ in the $i_{th}$ stage, there are three possible fusion situations: 1) when fusing features from the same branch, a simple identity connection is applied; 2) when features come from the above $a_{th}$ branch, *i.e.*, higher resolution than current one, $j_{th} - a_{th}$ number of successive downsampling modules are employed; 3) if features from the below $b_{th}$ branch, *i.e.*, lower resolution than current one, an upsampling module is adopted. Finally, those processed features are summed as the updated features for $j_{th}$ branch. For example, we can calculate the fused features of $2_{nd}$ branch in the $3_{rd}$ stage as follows:

$$\bar{\boldsymbol{F}}_{32} = \boldsymbol{F}_{32} + \text{downsampling}(\boldsymbol{F}_{31}) + \text{upsampling}(\boldsymbol{F}_{33}). \tag{8}$$

To effectively communicate different scales, the above fusion process are frequently made between and inside stages. However, those resampling operations, including both downsampling and upsampling,

are extremely time-consuming in point clouds due to the unordered characteristics. For example, when provided with an input point cloud containing $N$ points, the classical farthest point sampling (FPS) (Eldar et al., 1997; Qi et al., 2017b) requires the calculation of downsampling and upsampling neighboring indices. This process has a time complexity of $\mathcal{O}(N^2)$ as mentioned in Hu et al. (2020). Another option could be the simpler grid pooling and unpooling method proposed in Wu et al. (2022), which divides a point cloud into non-overlapping grids. The pooling is to sample each grid of points as a new points, and the unpooling is simply back-projecting the point to the original higher resolution grid of points.

However, both of them require calculating the indices for knn collection and resampling in each operation. This leads to a high computational cost, particularly because of the numerous resampling operations in PointHR. Fortunately, we identify those indices only depend on current point scale, which stays unchanged throughout the entire network. Specifically, we find that those knn indices solely rely on point coordinates so that each branch with the same resolution shares the same knn indices across all stages. As for resampling indices, they can be also shared when operating two specific resolution branches across different stages. Hence, we propose to *pre-compute* the indices for knn collection and resampling operation, which are saved to the cache to avoid on-the-fly computations, thus making it possible to efficiently employ high-resolution architectures for 3D dense point cloud analysis. The detailed description can be found in Appendix F.

## 4 EXPERIMENTS

### 4.1 DATASETS AND METRICS

We evaluate the proposed PointHR on two widely-used benchmarks, *i.e.*, S3DIS (Armeni et al., 2016) and ScanNetV2 (Dai et al., 2017), for point cloud semantic segmentation. S3DIS contains 271 rooms in 6 areas collected from three different buildings. Each point in the room is annotated with one of 13 semantic categories such as *"ceiling"* and *"bookcase"*. Following previous methods (Wu et al., 2022; Zhao et al., 2021), we keep Area 5 as the testing set and use remains for training PointHR. ScanNetV2 is another larger dataset, which consists of 1,513 room scenes, where 1,201 scenes for training and 312 for validation. Point clouds are created by sampling vertices from meshes that are reconstructed from RGB-D frames. Each sampled point is then assigned a semantic label from one of 20 categories, such as *"floor"* and *"table"*. Regarding the evaluation metrics, similar to Qian et al. (2022); Wu et al. (2022); Zhao et al. (2021), we adopt the mean class-wise intersection over union (mIoU), mean of class-wise accuracy (mAcc), and overall point-wise accuracy (OA). We further present evaluations on ShapeNetPart (Yi et al., 2016) and ModelNet40 (Wu et al., 2015) in Appendix B and C.

### 4.2 IMPLEMENTATION DETAILS

For the configurations of PointHR, unless otherwise stated, we employ $(M_1, M_2, M_3, M_4) = (1, 1, 5, 4)$ and $(C_1, C_2, C_3, C_4) = (64, 32, 32, 32)$, $B_i = 2$ and $K_i = 16$ for all $i \in \{0, 1, 2, 3\}$, the grouped vector attention defined by Equation 5 as the sequence operator, as well as the grid pooling and unpooling discussed in Section 3.3 as the sampling strategy.

For S3DIS (Armeni et al., 2016), following the practice in Wu et al. (2022); Zhao et al. (2021), the grid sampling with size 0.04m is first employed on the raw input points. During the training process, we apply popular data augmentations such flip, scale, jitter, random drop, and we also use the sphere crop on the entire scene and constrain the maximum number of input points to 100,000. Considering the training set of S3DIS is relatively small (*i.e.*, only 204 samples), we follow Qian et al. (2022); Wu et al. (2022); Zhao et al. (2021) to enlarge the size by repeating $30\times$ to obtain 6,120 samples. We train PointHR using four V100 GPUs for 100 epochs with batch size 12, and set the learning rate to 0.006 and drop it by $1/10$ at 60 and 80 epochs. AdamW optimizer (Loshchilov & Hutter, 2017) with the weight decay 0.05 and cross-entropy loss are applied. The pooling size are set to $(0.1, 0.2, 0.4, 0.8)$ for gird pooling to achieve approximately $6\times$ downsampling scale. For ScanNetV2 (Dai et al., 2017), we follow Wu et al. (2022) to employ grid sampling with size 0.02m on the raw point clouds. We also repeat the its training set (1,201 scenes) by $9\times$ to get 10,809 samples. Considering its larger scale than S3DIS, AdamW optimizer (Loshchilov & Hutter, 2017) with a smaller weight decay 0.02 are applied. OneCycleLR (Smith & Topin, 2019) scheduler is employed, where the learning raises from 0.0005 to 0.005 in the first 5 epochs and cosine annealing to 0 in the remaining 95 epochs.

Table 1: Quantitative results under mIoU (%), mAcc (%), and OA (%) metrics including per-category IoU are reported on Area 5 of S3DIS (Armeni et al., 2016). The **bold** denotes the best performance.

| Method | mIoU | mAcc | OA | ceiling | floor | wall | beam | column | window | door | table | chair | sofa | bookcase | board | clutter |
|---|---|---|---|---|---|---|---|---|---|---|---|---|---|---|---|---|
| PointNet (Qi et al., 2017a) | 41.1 | 49.0 | - | 88.8 | 97.3 | 69.8 | 0.1 | 3.9 | 46.3 | 10.8 | 59.0 | 52.6 | 5.9 | 40.3 | 26.4 | 33.2 |
| SegCloud (Tchapmi et al., 2017) | 48.9 | 57.4 | - | 90.1 | 96.1 | 69.9 | 0.0 | 18.4 | 38.4 | 23.1 | 70.4 | 75.9 | 40.9 | 58.4 | 13.0 | 41.6 |
| PointCNN (Li et al., 2018b) | 57.3 | 63.9 | 85.9 | 92.3 | 98.2 | 79.4 | 0.0 | 17.6 | 22.8 | 62.1 | 74.4 | 80.6 | 31.7 | 66.7 | 62.1 | 56.7 |
| PCT (Guo et al., 2021) | 61.3 | 67.7 | - | 92.5 | 98.4 | 80.6 | 0.0 | 19.4 | 61.6 | 48.0 | 76.6 | 85.2 | 46.2 | 67.7 | 67.9 | 52.3 |
| HPEIN (Jiang et al., 2019) | 61.9 | 68.3 | 87.2 | 91.5 | 98.2 | 81.4 | 0.0 | 23.3 | 65.3 | 40.0 | 75.5 | 87.7 | 58.5 | 67.8 | 65.6 | 49.4 |
| MinkUNet (Choe et al., 2019) | 65.4 | 71.7 | - | 91.8 | 98.7 | 86.2 | 0.0 | 34.1 | 48.9 | 62.4 | 81.6 | 89.8 | 47.2 | 74.9 | 74.4 | 58.6 |
| KPConv (Thomas et al., 2019) | 67.1 | 72.8 | - | 92.8 | 97.3 | 82.4 | 0.0 | 23.9 | 58.0 | 69.0 | 81.5 | 91.0 | 75.4 | 75.3 | 66.7 | 58.9 |
| CGA-Net (Lu et al., 2021) | 68.6 | - | - | 94.5 | 98.3 | 83.0 | 0.0 | 25.3 | 59.6 | 71.0 | 92.2 | 82.6 | 76.4 | 77.7 | 69.5 | 61.5 |
| CBL (Tang et al., 2022) | 69.4 | 75.2 | 90.6 | 93.9 | 98.4 | 84.2 | 0.0 | 37.0 | 57.7 | 71.9 | 91.7 | 81.8 | 77.8 | 75.6 | 69.1 | 62.9 |
| PTv1 (Zhao et al., 2021) | 70.4 | 76.5 | 90.8 | 94.0 | 98.5 | 86.3 | 0.0 | 38.0 | 63.4 | 74.3 | 89.1 | 82.4 | 74.3 | 80.2 | 76.0 | 59.3 |
| PointNext (Qian et al., 2022) | 70.5 | 76.8 | 90.6 | 94.2 | 98.5 | 84.4 | 0.0 | 37.7 | 59.3 | 74.0 | 83.1 | 91.6 | 77.4 | 77.2 | 78.8 | 60.6 |
| PointMeta (Lin et al., 2023) | 71.3 | - | 90.8 | - | - | - | - | - | - | - | - | - | - | - | - | - |
| PointMixer (Choe et al., 2022) | 71.4 | 77.4 | - | 94.2 | 98.2 | 86.0 | 0.0 | 43.8 | 62.1 | 78.5 | 90.6 | 82.2 | 73.9 | 79.8 | 78.5 | 59.4 |
| PTv2 (Wu et al., 2022) | 71.6 | 77.9 | 91.1 | - | - | - | - | - | - | - | - | - | - | - | - | - |
| StraFormer (Lai et al., 2022) | 72.0 | 78.1 | 91.5 | 96.2 | 98.7 | 85.6 | 0.0 | 46.1 | 60.0 | 76.8 | 92.6 | 84.5 | 77.8 | 75.2 | 78.1 | 64.0 |
| PointHR (ours) | **73.2** | **78.7** | **91.8** | 94.0 | 98.5 | 87.5 | 0.0 | 53.7 | 62.9 | 80.2 | 84.2 | 92.5 | 75.4 | 76.5 | 84.8 | 61.8 |

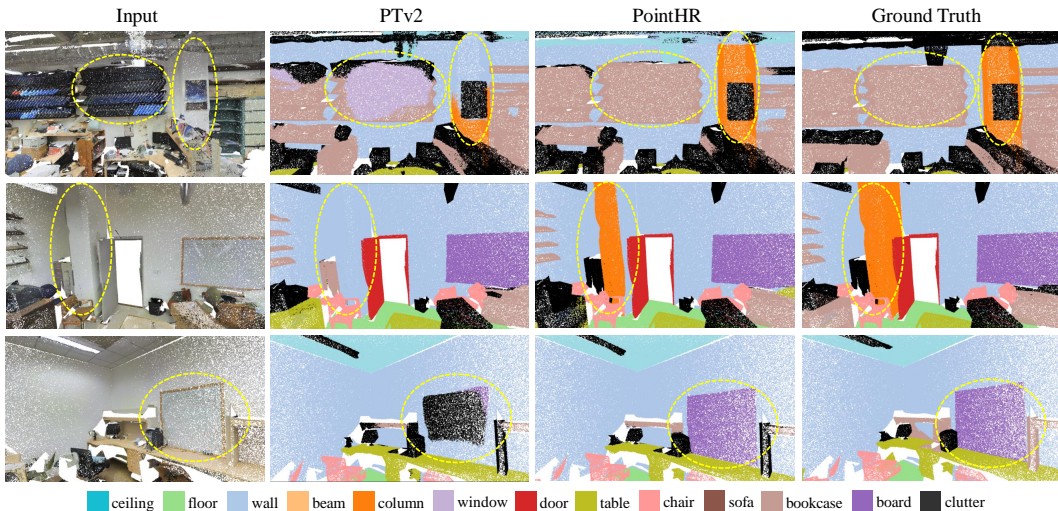

Figure 3: Visualization of point cloud segmentation results on the Area 5 of S3DIS. Note that yellow circles highlight the improvements made by PointHR over PTv2.

We use $(0.06, 0.15, 0.375, 0.9375)$ for gird pooling to approximate $6\times$ downsampling scale. Other settings are kept the same as in S3DIS.

## 4.3 RESULTS ON S3DIS

**Quantitative Results**: Table 1 demonstrates the results of recent state-of-the-art methods and the proposed PointHR on the Area 5 of S3DIS. Apparently, our PointHR achieves best performances on all three metrics, *i.e.*, mIoU (%), mAcc (%), and OA (%). It surpasses different kinds of methods including transformer-based (Lai et al., 2022; Wu et al., 2022; Zhao et al., 2021), mlp-based (Qian et al., 2022; Lin et al., 2023; Choe et al., 2022), and graph-based (Thomas et al., 2019; Li et al., 2018b) approaches. For example, PointHR outperforms the current state-of-the-art model, StraFormer (Lai et al., 2022), with a clear margin in terms of mIoU, 73.2% *vs.* 72.0%. It is also worthy noting that PointHR uses the same block (grouped vector attention) as in PTv2 (Wu et al., 2022), but PointHR takes the lead on all three metrics, *i.e.*, 73.2% *vs.* 71.6%, 78.7% *vs.* 77.9%, and 91.8% *vs.* 91.1%. We believe that this is partially because of the high-resolution architectures, which explicitly maintain multi-scale features in parallel and allow for interactions across scales. Besides, as for per-category IoU, we find that PointHR achieves better performances on those large flat objects such as *"column"*, *"door"*, and *"board"*. We thus conjecture that PointHR has a very good ability to efficiently capture information at different scales to accurately segment both boundary and inner points.

**Qualitative Results**: We visually compare the predictions made by PTv2 (Wu et al., 2022) and PointHR, as well as ground truths on Area 5 of S3DIS in Figure 3. As we see that the *"column"* area is challenging since it usually looks very similar to the *"wall"* area, but different only in shape. However, while both the *"column"* and *"wall"* have a flat shape in short-range views, they exhibit distinct characteristics in long-range perspectives. Specifically, the *"column"* takes on a cube-like shape, which serves as a key feature for differentiation from the *"wall"*. Our PointHR can effectively maintain high-resolution features and incorporate cross-scale fusion within the network architecture. This enables better capture long-long range contexts, which are crucial for accurate recognition on both internal and boundary points of *"columns"*. Similar situations are observed on the *"board"*.

## 4.4 RESULTS ON SCANNETV2

**Quantitative Results**: Next we evaluate PointHR on the more challenging benchmark Scan-NetV2. Similar to Han et al. (2020); Liu et al. (2021); Wu et al. (2022); Yang et al. (2023), except for the results on validation set, we have further sub-mited our predictions on testing set to the official testing server. All the performances using mIoU metric are reported in Table 2. Not surprisingly, PointHR delivers another state-of-the-art performance 76.6%, which surpasses the current best

Table 2: Quantitative results on ScanNetV2.

| Method | #Params | FLOPs | Val (%) | Test (%) |
|---|---|---|---|---|
| PointNet++ (Qi et al., 2017b) | 1.0M | 7.2G | 53.5 | 55.7 |
| RandLA-Net (Hu et al., 2020) | 1.3M | 5.8G | - | 64.5 |
| PointConv (Wu et al., 2019) | - | - | 61.0 | 66.6 |
| PointASNL (Yan et al., 2020) | - | - | 63.5 | 66.6 |
| KPConv (Thomas et al., 2019) | 15M | - | 69.2 | 68.6 |
| CBL (Tang et al., 2022) | 18.6M | - | - | 70.5 |
| PTv1 (Zhao et al., 2021) | 7.8M | 5.7G | 70.6 | - |
| PointNext (Qian et al., 2022) | 41.6M | 84.8G | 71.5 | 71.2 |
| PointMeta (Lin et al., 2023) | 19.7M | 11.0 | 72.8 | 71.4 |
| SparseCNN (Graham et al., 2018) | - | - | 69.3 | 72.5 |
| MinkUNet (Choy et al., 2019) | - | - | 72.2 | 73.6 |
| StraFormer (Lai et al., 2022) | 8.02M | 12.4G | 74.3 | 74.7 |
| BPNet (Hu et al., 2021) | - | - | 73.9 | 74.9 |
| PTv2 (Wu et al., 2022) | 11.3M | 14.3G | 75.4 | 75.2 |
| PointHR (ours) | 7.1M | 10.3G | 75.4 | **76.6** |

method, *i.e.*, 75.2% from PTv2 (Wu et al., 2022) even with about 40% fewer parameters and FLOPs (7.1M *vs.* 11.3M and 10.3G *vs.* 14.3G). In addition, PointHR directly taking points as input also outperforms those voxel-input methods (Hu et al., 2021; Choy et al., 2019) that can conveniently use 3D convolutions. Note that BPNet (Hu et al., 2021) achieves 74.9% using both point clouds and corresponding 2D images as input, which is still inferior to PointHR that only works on points.

**Qualitative Results**: Figure 4 demonstrates the semantic segmentation results obtained by PointHR on ScanNetV2. Our PointHR performs well on different kinds of scenes such as office, bathroom, and bedroom. It is also worth noting that PointHR can handle numerous objects in a scene well, as observed with the many chairs surrounding the table being precisely segmented.

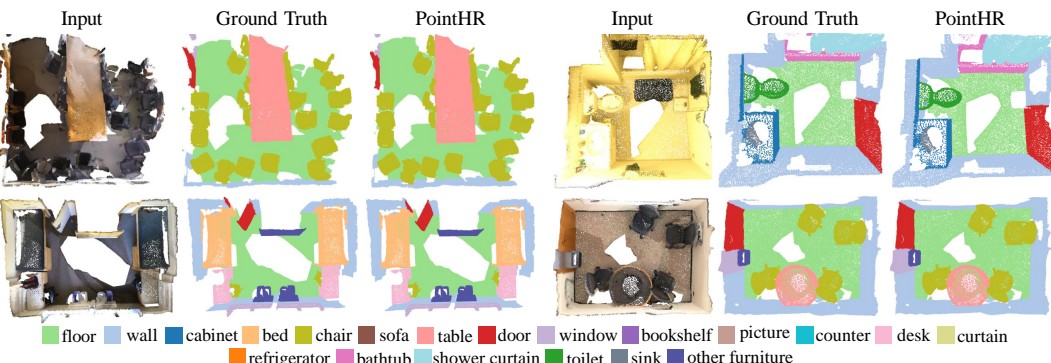

Figure 4: Visualization of point cloud segmentation by the proposed PointHR on ScanNetV2. More illustrations are available in Appendix H.

## 4.5 ABLATION STUDIES

All ablation studies are conducted on the validation set of ScanNetV2.

Table 3: Ablation studies on model scalability.

| Model | $(M_1, \cdots, M_4)$ | $(C_1, \cdots C_4)$ | #Params | FLOPs | mIoU (%) |
|---|---|---|---|---|---|
| PointHR-T | $(1, 1, 2, 1)$ | $(64, 16, 16, 16)$ | 0.6M | 2.5G | 72.7 |
| PointHR-S | $(1, 1, 3, 2)$ | $(64, 16, 16, 16)$ | 1.0M | 2.8G | 73.0 |
| PointHR-B | $(1, 1, 3, 2)$ | $(64, 32, 32, 32)$ | 3.9M | 6.8G | 74.9 |
| PointHR-L | $(1, 1, 5, 4)$ | $(64, 32, 32, 32)$ | 7.1M | 10.3G | 75.4 |

**Model Scalability**: We investigate the scalability of PointHR by fixing $B_i = 2$, $K_i = 16$ for all $i \in \{0, 1, 2, 3\}$ and increasing modules $M = (M_1, M_2, M_3, M_4)$ and channels $C = (C_1, C_2, C_3, C_4)$. Four different scales of PointHR are obtained as demonstrated in Table 3. As we can observe, the overall trend is that the metric mIoU increase as the model size grows. PointHR-S only marginally outperforms PointHR-T with deeper depth, which we suspect is because only 16 channels for $\{C_i | i \in \{1, 2, 3\}\}$ significantly limit the capacity of model. When doubling the channels to 32 as PointHR-B, the performance is remarkably boosted from 73.0% to 74.9%. We further increase the depth to get PointHR-L with the best performance 75.4%.

**Sequence Operator**: We explore three different sequence operators discussed in Section 3.2, *i.e.*, pure MLPs (Lin et al., 2023), vector attention (Zhao et al., 2021), and grouped vector attention (Wu et al., 2022) defined by Equation 6, 4, and 5 respectively. The results are presented in Table 4 by comparing PointHR with the specific sequence operator with its original method. As we can see from the first group, PointHR-B integrated with *mlps* surpasses PointMeta (Lin et al., 2023) by 1.4% mIoU but with significantly smaller parameters and FLOPs (2.3M *vs.* 19.7M and 1.8G *vs.* 11.0G) . Meanwhile, PointHR-B with *va* improves the performance of PTv1 (Zhao et al., 2021) by a large margin 4% with approximately half parameters and FLOPs. The above observations confirm the effectiveness and generalization ability of PointHR.

**Resampling Strategy**: We compare the furthest point sampling plus KNN to grid pooling under the model configuration PointHR-B. The FPS version named as PointHR-B-FPS is also *pre-computing* the downsampling and upsampling index for later fetching to make a fair comparison. It achieves 73.9% mIoU, which is clearly inferior to 74.9% made by PointHR-B. Besides, the speed of PointHR is about 40% faster than PointHR-B-FPS as it requires 244 V100 GPU hours while PointHR-B-FPS needs 412 GPU hours. If the model becomes deeper and increases the frequency of cross scale interactions, the gap on speed will further grow.

Table 4: Ablation studies on different sequence operators. *mlps*: pure MLPs (Lin et al., 2023), *va*: vector attention (Zhao et al., 2021), *gva*: grouped vector attention (Wu et al., 2022). HR denotes the high-resolution architecture.

| Method | Sequence Operator | HR | #Params | FLOPs | Val (%) | Test (%) |
|---|---|---|---|---|---|---|
| PointMeta (Lin et al., 2023) | *mlps* | ✗ | 19.7M | 11.0G | 72.8 | 71.4 |
| PointHR-B | *mlps* | ✓ | 2.3M | 1.8G | **74.2** | - |
| PTv1 (Zhao et al., 2021) | *va* | ✗ | 7.8M | 5.7G | 70.6 | - |
| PointHR-B | *va* | ✓ | 3.4M | 3.0G | **74.6** | - |
| PTv2 (Wu et al., 2022) | *gva* | ✗ | 11.3M | 14.3G | **75.4** | 75.2 |
| PointHR-B | *gva* | ✓ | 3.9M | 6.8G | 74.9 | - |
| PointHR-L | *gva* | ✓ | 7.1M | 10.3G | **75.4** | **76.6** |

## 5 CONCLUSION

In this paper, we explore high-resolution architectures for 3D point cloud segmentation. To achieve this, we formulate the proposed PointHR in a unified way using a sequence operator and a resampling operator, enabling use of those off-the-shelf point cloud blocks and modules without additional efforts. Besides, we propose to *pre-compute* the indices for knn collection and resampling operation to avoid on-the-fly computations, thus efficiently employing high-resolution architectures. Comprehensive experiments on popular point cloud segmentaion datasets, S3DIS (Armeni et al., 2016) and Scan-NetV2 (Dai et al., 2017), demonstrate the effectiveness of high-resolution architectures for 3D dense point cloud analysis and also yield new state-of-the-art performances.

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

## APPENDIX

In this appendix, we begin by presenting the typical configurations table of PointHR. Next, we evaluate the proposed PointHR on ShapeNetPart (Yi et al., 2016) and ModelNet40 (Wu et al., 2015), respectively. Then we conduct ablation studies on the decoder design. Subsequently, the per-category IoUs on the testing split of ScanNetV2 are provided. After that, we demonstrate the details of *pre-compute* indices as outlined in Section 3.3. Finally, we deliver more additional visualizations.

## A  CONFIGURATIONS

Table 5: **The typical PointHR configuration. SO**: sequence operator; $S$: scale factor; $M_i$: the number of modules; $B_i$: the number of blocks; $K_i$: the number of neighbors; $C_i$: the number of channels.

| Res. | Stage 1 | Stage 2 | Stage 3 | Stage 4 |
|---|---|---|---|---|
| $S\times$ | $[ \ \mathbf{SO}, K_1, C_1 \ ]\times B_1 \times M_1$ | $[ \ \mathbf{SO}, K_2, C_2 \ ]\times B_2 \times M_2$ | $[ \ \mathbf{SO}, K_3, C_3 \ ]\times B_3 \times M_3$ | $[ \ \mathbf{SO}, K_4, C_4 \ ]\times B_4 \times M_4$ |
| $S^2\times$ | | $[ \ \mathbf{SO}, K_2, 2C_2 \ ]\times B_2 \times M_2$ | $[ \ \mathbf{SO}, K_3, 2C_3 \ ]\times B_3 \times M_3$ | $[ \ \mathbf{SO}, K_4, 2C_4 \ ]\times B_4 \times M_4$ |
| $S^3\times$ | | | $[ \ \mathbf{SO}, K_3, 4C_3 \ ]\times B_3 \times M_3$ | $[ \ \mathbf{SO}, K_4, 4C_4 \ ]\times B_4 \times M_4$ |
| $S^4\times$ | | | | $[ \ \mathbf{SO}, K_4, 8C_4 \ ]\times B_4 \times M_4$ |

## B  EVALUATION ON SHAPENETPART

We have further evaluated the proposed PointHR on ShapeNetPart (Yi et al., 2016), which stands as a widely recognized dataset employed for the task of point cloud part segmentation. This dataset comprises 16,880 3D models, categorized into 16 distinct shape categories, such as *"car"* and *"table"*. In the context of data splitting, 14,006 models are designated for the training set, while the remaining 2,874 models are reserved for the testing set. Notably, each category exhibits a varying number of constituent parts, ranging from 2 to 6, yielding a total of 50 distinct parts across all categories. The reported evaluation metric is the instance mean Intersection over Union (mIoU). Table 6 presents the results of PointHR compared to previous methods. As observed, our PointHR achieves state-of-the-art performance $87.2\%$ with reasonable parameters 7.4M.

Table 6: Results on ShapeNetPart.

| Method | #Params | mIoU (%) |
|---|---|---|
| PointNet (Qi et al., 2017a) | 3.6M | 83.7 |
| PointNet++ (Qi et al., 2017b) | 1.5M | 85.1 |
| DGCNN (Wang et al., 2019) | 1.3M | 85.2 |
| PointConv (Wu et al., 2019) | - | 85.7 |
| ASSANet-L (Qian et al., 2021) | - | 86.1 |
| PointMLP (Ma et al., 2022) | 13.2M | 86.1 |
| PVCNN (Liu et al., 2019) | - | 86.2 |
| PCT (Guo et al., 2021) | - | 86.4 |
| KPConv (Thomas et al., 2019) | 14.3M | 86.4 |
| PTv1 (Zhao et al., 2021) | 7.8M | 86.6 |
| StraFormer (Lai et al., 2022) | 8.0M | 86.6 |
| CurveNet (Xiang et al., 2021) | - | 86.8 |
| PointNeXt (Qian et al., 2022) | 22.5M | 87.0 |
| PointHR (ours) | 7.4M | **87.2** |

Table 7: Results on ModelNet40.

| Method | Inputs | #Points | OA(%) |
|---|---|---|---|
| PointNet (Qi et al., 2017a) | xyz | 1024 | 89.2 |
| PointNet++ (Qi et al., 2017b) | xyz | 1024 | 90.7 |
| PointNet++ (Qi et al., 2017b) | xyz+norm | 5000 | 91.9 |
| PointCNN (Li et al., 2018b) | xyz | 1024 | 92.5 |
| PointConv (Wu et al., 2019) | xyz+norm | 1024 | 92.5 |
| KPConv (Thomas et al., 2019) | xyz | 7000 | 92.9 |
| DGCNN (Wang et al., 2019) | xyz | 1024 | 92.9 |
| PointASNL (Yan et al., 2020) | xyz | 1024 | 92.9 |
| PointNext (Qian et al., 2022) | xyz | 1024 | 93.2 |
| PosPool (Liu et al., 2020) | xyz | 5000 | 93.2 |
| PCT (Guo et al., 2021) | xyz | 1024 | 93.2 |
| SO-Net (Li et al., 2018a) | xyz | 5000 | 93.4 |
| PTv1 (Zhao et al., 2021) | xyz | 1024 | 93.7 |
| PointMLP (Ma et al., 2022) | xyz | 1024 | **94.1** |
| PointHR (ours) | xyz | 1024 | 93.9 |

## C   EVALUATION ON MODELNET40

Although PointHR is specifically designed for point cloud segmentation, it can also be used for classification tasks with slight modifications. Specifically, we modify the feature propagation direction in the decoder by changing it from low-to-high to high-to-low, and then applying a global maxpooling. We evaluate PointHR on ModelNet40 dataset (Wu et al., 2015), which serves as a canonical dataset widely utilized for point cloud classification. This dataset comprises a total of 9,843 CAD models allocated to the training set, with an additional 2,468 CAD models designated for the testing set. These models span across 40 distinct object categories. The primary evaluation metric employed for assessing model performance is the overall accuracy (OA). We report the performance of PointHR and other previous approaches in Table 7. Finally, PointHR achieves comparable accuracy to previous state-of-the-art methods.

## D   DECODER DESIGN

For the decoder design, we conduct ablation studies including 1) directly sum of different resolution features; 2) progressively fusion of adjacent features; and 3) progressively fusion of adjacent features with the sequence operator for refinement, which are corresponding to the Sum, PG and PGR of Table 8, respectively. We choose PointHR-T as the baseline and perform corresponding experiments on ScanNet under mIoU (%). As shown in Table 8, we find that progressively fusing adjacent features brings high performance, which can be further enhanced by a following sequence operator for refinement. Hence, we opt for the final decoder design as the default.

Table 8: Strategies.

| | PointHR-T |
|---|---|
| Sum | 71.6% |
| PG | 71.9% |
| PGR | 72.7% |

Table 9: Results of per-category IoU (%) on the testing set from ScanNetV2 corresponding to Table 2.

| Method | mIoU | bathtub | bed | bookshelf | cabinet | chair | counter | curtain | desk | door | floor | other furniture | picture | refrigerator | shower curtain | sink | sofa | table | toilet | wall | window |
|---|---|---|---|---|---|---|---|---|---|---|---|---|---|---|---|---|---|---|---|---|---|
| PointNet++ | 55.7 | 73.5 | 66.1 | 68.6 | 49.1 | 74.4 | 39.2 | 53.9 | 45.1 | 37.5 | 94.6 | 37.6 | 20.5 | 40.3 | 35.6 | 55.3 | 64.3 | 49.7 | 82.4 | 75.6 | 51.5 |
| RandLA-Net | 64.5 | 77.8 | 73.1 | 69.9 | 57.7 | 82.9 | 44.6 | 73.6 | 47.7 | 52.3 | 94.5 | 45.4 | 26.9 | 48.4 | 74.9 | 61.8 | 73.8 | 59.9 | 82.7 | 79.2 | 62.1 |
| PointConv | 66.6 | 78.1 | 75.9 | 69.9 | 64.4 | 82.2 | 47.5 | 77.9 | 56.4 | 50.4 | 95.3 | 42.8 | 20.3 | 58.6 | 75.4 | 66.1 | 75.3 | 58.8 | 90.2 | 81.3 | 64.2 |
| PointASNL | 66.6 | 70.3 | 78.1 | 75.1 | 65.5 | 83.0 | 47.1 | 76.9 | 47.4 | 53.7 | 95.1 | 47.5 | 27.9 | 63.5 | 69.8 | 67.5 | 75.1 | 55.3 | 81.6 | 80.6 | 70.3 |
| KPConv | 68.4 | 84.7 | 75.8 | 78.4 | 64.7 | 81.4 | 47.3 | 77.2 | 60.5 | 59.4 | 93.5 | 45.0 | 18.1 | 58.7 | 80.5 | 69.0 | 78.5 | 61.4 | 88.2 | 81.9 | 63.2 |
| CBL | 70.5 | 76.9 | 77.5 | 80.9 | 68.7 | 82.0 | 43.9 | 81.2 | 66.1 | 59.1 | 94.5 | 51.5 | 17.1 | 63.3 | 85.6 | 72.0 | 79.6 | 66.8 | 88.9 | 84.7 | 68.9 |
| PointNext | 71.2 | - | - | - | - | - | - | - | - | - | - | - | - | - | - | - | - | - | - | - | - |
| PointMeta | 71.4 | 83.5 | 78.5 | 82.1 | 68.4 | 84.6 | 53.1 | 86.5 | 61.4 | 59.6 | 95.3 | 50.0 | 24.6 | 67.4 | 88.8 | 69.2 | 76.4 | 62.4 | 84.9 | 84.4 | 67.5 |
| SparseCNN | 72.5 | 64.7 | 82.1 | 84.6 | 72.1 | 86.9 | 53.3 | 75.4 | 60.3 | 61.4 | 95.5 | 57.2 | 32.5 | 71.0 | 87.0 | 72.4 | 82.3 | 62.8 | 93.4 | 86.5 | 68.3 |
| MinkUNet | 73.6 | 85.9 | 81.8 | 83.2 | 70.9 | 84.0 | 52.1 | 85.3 | 66.0 | 64.3 | 95.1 | 54.4 | 28.6 | 73.1 | 89.3 | 67.5 | 77.2 | 68.3 | 87.4 | 85.2 | 72.7 |
| StraFormer | 74.7 | 90.1 | 80.3 | 84.5 | 75.7 | 84.6 | 51.2 | 82.5 | 69.6 | 64.5 | 95.6 | 57.6 | 26.2 | 74.4 | 86.1 | 74.2 | 77.0 | 70.5 | 89.9 | 86.0 | 73.4 |
| BPNet | 74.9 | 90.9 | 81.8 | 81.1 | 75.2 | 83.9 | 48.5 | 84.2 | 67.3 | 64.4 | 95.7 | 52.8 | 30.5 | 77.3 | 85.9 | 78.8 | 81.8 | 69.3 | 91.6 | 85.6 | 72.3 |
| PTv2 | 75.2 | 74.2 | 80.9 | 87.2 | 75.8 | 86.0 | 55.2 | 89.1 | 61.0 | 68.7 | 96.0 | 55.9 | 30.4 | 76.6 | 92.6 | 76.7 | 79.7 | 64.4 | 94.2 | 87.6 | 72.2 |
| PointHR | 76.6 | 79.0 | 82.3 | 88.1 | 74.9 | 87.1 | 58.7 | 91.8 | 65.5 | 68.5 | 97.3 | 56.0 | 36.3 | 58.2 | 93.3 | 81.6 | 82.7 | 69.8 | 97.4 | 89.7 | 73.9 |

## E    PER-CATEGORY IoU ON SCANNETV2

We additionally provide the results of per-category IoU (%) on the testing split of ScanNetV2 to complement the previous Table 2. All the results are obtained from the official testing leaderboard[1] and demonstrated in Table 9.

## F    PRE-COMPUTE INDICES

As discussed in Section 3.3, we propose to *pre-compute* the indices for knn collection and resampling operation, which are saved to the cache to avoid on-the-fly computations. Here, we provide an example to illustrate how to *pre-compute* the resampling indices.

Taking the FPS with KNN resampling strategy as an example, we need to compute the index mapping high-resolution to low-resolution for downsampling and the index of $K$ neighbors that interpolate low-resolution to high-resolution for upsampling. Recall that these down- and upsampling operations are abundant in PointHR. However, we found that the resampling operations in the later stages includes those in the previous stages. For example, the index for downsampling from the $2_{nd}$ to the $3_{rd}$ branch in stage 3 is the same as the $2_{nd}$ to $3_{rd}$ branch in stage 4 because they share the same resolutions. As such, we can compute all the mapping index, including both downsampling and upsampling index, in stage 4, which covers resampling relationships of all the stages. Specifically, in each iteration, we first downsample the input point clouds into four different resolutions and calculate all the downsampling and upsampling indices between these four resolutions. Thereafter, we feed the point cloud features, along with the index, to the network. This enables resampling operations to fetch the corresponding index to obtain down- and up-scale features, thereby speeding up the whole process. We compared the training latency of one iteration with 3 batch size in one V100 GPU for PointHR model with and without the *precomputed* neighbor index. The results showed a latency of 1.86s and 2.18s, respectively. Note that 100 epochs consist of approximately 360,000 iterations, indicating that the *precomputed* neighbor index can save about 32 GPU hours for one training process. A similar situation arises for employing grid pooling and unpooling.

## G    MEMORY ANALYSIS

Since our PointHR maintains high-resolution branch throughout all the stage, the memory usage would be a concern. However, two designs of PointHR significantly reduce the heavy memory cost. The first one is that the highest resolution branch across all stage is $N/S$ instead of $N$ as illustrated in Figure 1, where $S = 6$ in our experiments. The second strategy is that we employ a small beginning feature dimension, *e.g.*, the channel of the highest resolution is only 32 for PointHR-L as shown in Table 3. Here we provide a quantitative comparison on memory

Table 10: Comparisons on memory usage (G).

| Method | Memory(G) |
|---|---|
| PointNext (Qian et al., 2022) | 43.14 |
| PointMeta (Lin et al., 2023) | 20.38 |
| PTv1 (Zhao et al., 2021) | 15.13 |
| PTv2 (Wu et al., 2022) | 21.50 |
| PointHR-T | 9.40 |
| PointHR-S | 10.84 |
| PointHR-B | 17.53 |
| PointHR-L | 23.58 |

usage between PointHR and recent state-of-the-art methods (Qian et al., 2022; Lin et al., 2023; Zhao et al., 2021; Wu et al., 2022) using 300K points as an example for input on a single GPU (because both PTv2 (Wu et al., 2022) and PointHR use a batch of three point clouds with 100K points per GPU). As shown in the Table 10, all the different configurations of PointHR achieve the reasonable memory usage, *e.g.*, the largest variant PointHR-L can be trained on the 24GB GPU.

## H    VISUALIZATIONS

More visualizations on S3DIS and ScanNetV2 are illustrated in Figure 5 and 6. Thanks to its multi-scale characteristic, the proposed PointHR is capable of predicting accurate semantic categories for point clouds even on challenging scenes. For example, PointHR segments the difficult boundaries

---

[1]https://kaldir.vc.in.tum.de/scannet_benchmark/semantic_label_3d

such as legs of chairs and tables precisely, and also makes smooth predictions on the inner area of large objects like board.

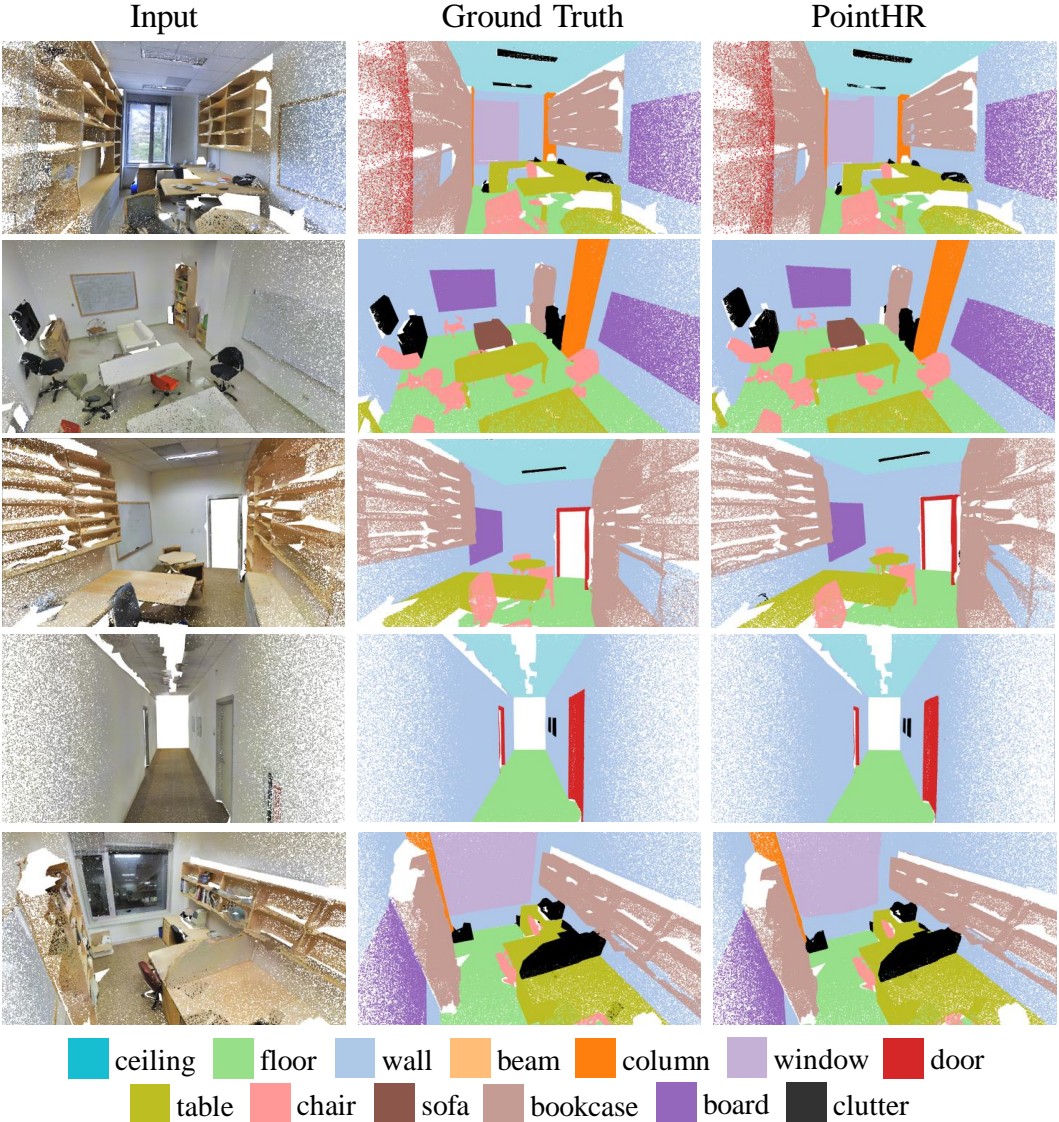

Figure 5: Visualizations of point cloud segmentation by PointHR on S3DIS. From left to right: input, ground truth, and predictions made by PointHR.

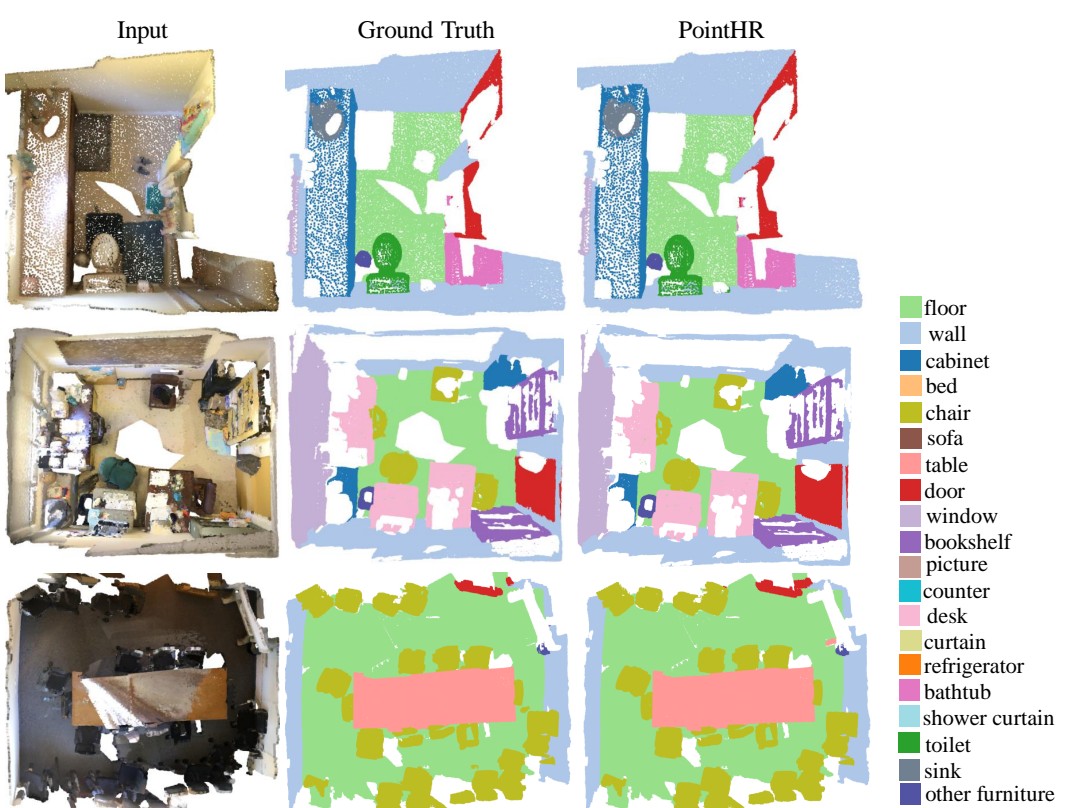

Figure 6: Visualizations of point cloud segmentation by PointHR on ScanNetV2

