# OpenReview forum: "PointHR: Exploring High-Resolution Architectures for 3D Point Cloud Segmentation"
_ICLR.cc/2024/Conference — ICLR 2024 Conference Withdrawn Submission_

### Official Review · Reviewer_MgXg · 2023-10-29

**Soundness:** 3 good
**Presentation:** 3 good
**Contribution:** 3 good
**Rating:** 6
**Confidence:** 5

**Summary:**

This paper extends the idea from HRNet of 2D image understanding into 3D point cloud segmentation, yielding a new transformer-based network called PointHR. The key of PointHR is to maintain high-resolution through the whole network, which benefits the dense prediction task. Such design is employed on pure MLPs and efficient attention operators, to formulate a knn-based sequence operator. For verifying the effectiveness of HRNet, experiments are conducted on S3DIS and ScanNet v2 for scene segmentation.

**Strengths:**

1. The paper is well-organized, well-written, and easy to follow.
2. Although the proposed method relies on many existing methods such as point MLP or attention operators, point resampling strategies, the proposed method nicely analyzes, utilizes, and fuses them into a complete pipeline without complicated additional effort, which is straightforward, technically sound, and effective.
3. The proposed method achieves superior results on S3DIS Area-5 and ScanNetv2.
4. It is good to see that in Table 4, even using pure MLPs, PointHR can also achieve good results with decent efficiency.
5. #Params and FLOPs are also listed and compared with other methods, which is meaningful for verifying the efficiency of the proposed PointHR.
6. Model scalability is explored in Table 3, which is important for the era of large models.
7. The code implementations are provided in the supplementary material and discussed in the paper.
8. The supplementary material provides more result visualizations and states the possible limitations.
9. The memory usage is also compared.

**Weaknesses:**

A representative of point convolution methods, PAConv [1] should be discussed and compared.
[1] PAConv: Position Adaptive Convolution with Dynamic Kernel Assembling on Point Clouds. CVPR, 2021.


**Additional comments**:
1. It would be better if some results on the point cloud classification task could be provided in the supplementary material, to fully explore the proposed method.
2. Stratified Transformer provides the robustness analysis, it is recommended to test the robustness of the proposed PointHR, which is important for real-world applications.

**Questions:**

This work shows great performance and could be a new backbone for 3D point cloud segmentation. Hope the code and the pretrained models can be public upon the acceptance of this paper.

However, the paper mentioned in the weakness part should be discussed and compared. A good literature review is also highly important for a good paper.

Moreover, considering the blossoming of fully-supervised methods in recent years, the new direction of 3D point cloud processing may go to the exploration of self/un-supervised, open-vocabulary, and open-domain methods at the next stage, this paper does not contribute **exceptional** value to 3D point cloud segmentation. Thus, the rating is 6 instead of 8.

---

### Official Review · Reviewer_PsJ9 · 2023-11-01

**Soundness:** 3 good
**Presentation:** 3 good
**Contribution:** 2 fair
**Rating:** 3
**Confidence:** 5

**Summary:**

The paper presents PointHR, which is a high-resolution point cloud semantic segmentation framework. PointHR extends the HRNet (High-Resolution Network) in 2D to 3D point cloud processing and explores different sequence operators (e.g., MLPs, vector attention, grouped vector attention) and resampling operators (e.g., KNN & grid pooling) in the framework. The paper shows good performance on ScanNet and S3DIS dataset.

**Strengths:**

The paper provides detailed results of applying different point sequence operations and sampling strategies in a high-resolution-style network. The experiments show that with lower FLOPs, the model can achieve better performance than the original downsampling-upsampling (encoder-decoder)-style network design.

**Weaknesses:**

1. The paper seems to be the application of previously proposed point cloud operations such as MLP and self-attention in high-resolution networks (proposed before in 2D), thus lacking a demonstrable level of novelty.
2. KNN in the point sequence operator is also time-consuming especially when the point number is large. Some methods like MinkowskiNet (4D Spatio-Temporal ConvNets: Minkowski Convolutional Neural Networks) and OctFormer (OctFormer: Octree-based Transformers for 3D Point Clouds) also avoid the usage of KNN. The comparison of the FLOPs and performance with these methods is also expected.
3. The inference time comparison is needed as the inference time cannot be totally inferred from the FLOPs considering operations like KNN and FPS in point cloud processing.
3. There are some typos in the paper

**Questions:**

Please refer to the weaknesses.

There are also some typos in the paper. For example:
- In the first paragraph of Sec. 3.1, N, cls -> N \times cls
- the arrow and channel number in the second stage in figure 1

---

### Official Review · Reviewer_BBdx · 2023-11-09

**Soundness:** 3 good
**Presentation:** 4 excellent
**Contribution:** 2 fair
**Rating:** 3
**Confidence:** 5

**Summary:**

This paper focuses on the problem of 3D point cloud segmentation. The author proposes a framework named PointHR, which adapts the 2D HRNet architecture to the 3D domain, enabling the learning of semantically rich and spatially precise point cloud representations at high resolutions. The experimental evaluations are conducted on several point cloud segmentation datasets and the experimental results look good. It achieves comparable or favorable performance to other high-performance solutions like PTv2, while with less number of network parameters.

**Strengths:**

1. The experimental evaluations are conducted on two popular and challenging datasets S3DIS and ScanNetv2, and the proposed solution shows better performance than the previous methods.
2. The presentation is clear and easy to follow, in total, the content is well organized and presented. Utilizing diagrams, charts, and additional visual elements not only improves the paper's comprehensibility but also broadens its appeal to a more diverse audience.

**Weaknesses:**

Overall, the contribution of this paper is insufficient and may not provide valuable insights into this research field. The main concern of the paper is its technical novelty.
1. The proposed PointHR appears to be a simple combination of the existing 3D point cloud segmentation algorithm PTv2 and the popular 2D backbone structure HRNet. The overall structure follows a similar design of HRNet, as shown in Figure 1, the illustration is very similar to that in HRNet.
2. The basic raw operator adopts previous designs as in Figure 2, moreover, the formulation of equation 5 is also quite similar to that in PTv2. Though the results seem better than previous solutions, a simple combination of two existing solutions might not motivate potential readers much.
3. Further in-depth analysis and core designs are needed to tackle the specific core challenges in 3D scene understanding, thus making the paper and solution more appealing and inspiring to potential readers.

**Questions:**

The key challenges in 3D point cloud segmentation and what kinds of new designs are needed to tackle these challenges, methods need to be more appealing and inspiring, thus motivating potential researchers.